# Definition of Meat Quality Across Different Cattle Breeds

**DOI:** 10.3390/ani15233467

**Published:** 2025-12-02

**Authors:** Beniamino T. Cenci-Goga, Egidia Costanzi, Francesca Blasi, Federica Ianni, Marco Tassinari, Claudio Truzzi, Musafiri Karama, Saeed El-Ashram, Cristina Saraiva, Marcelo Martínez-Barbitta, Juan García-Díez, Massimo Zerani, Gabriella Guelfi, Margherita Maranesi, Luca Grispoldi, Lina Cossignani

**Affiliations:** 1Department of Veterinary Medicine, University of Perugia, 06126 Perugia, Italy; egidia.costanzi@unipg.it (E.C.); massimo.zerani@unipg.it (M.Z.); gabriella.guelfi@unipg.it (G.G.); margherita.maranesi@unipg.it (M.M.); grisluca@outlook.it (L.G.); 2Department of Paraclinical Sciences, Faculty of Veterinary Science, University of Pretoria, Onderstepoort, Pretoria 0110, South Africa; musafiri.karama@up.ac.za; 3Department of Pharmaceutical Sciences, University of Perugia, 06126 Perugia, Italy; francesca.blasi@unipg.it (F.B.); federica.ianni@unipg.it (F.I.); lina.cossignani@unipg.it (L.C.); 4Department of Veterinary Medical Sciences, University of Bologna, 40126 Bologna, Italy; marco.tassinari@unibo.it; 5METRO Italia S.p.A., 20097 San Donato Milanese, Italy; claudio.truzzi@metro.it; 6Department of Zoology, Faculty of Science, Kafrelsheikh University, Kafr El-Sheikh 33516, Egypt; saeed_elashram@yahoo.com; 7Department of Veterinary Sciences, School of Agrarian and Veterinary Sciences, University of Trás-os-Montes e Alto Douro, 5001-801 Vila Real, Portugal; crisarai@utad.pt (C.S.); juangarciadiez@utad.pt (J.G.-D.); 8Sistema Reproductivo Veterinario Integral Uruguay, Luis Alberto de Herrera 1154, Nueva Helvecia 70300, Uruguay; srvi.uy@gmail.com; 9Dottorato in Sanità e Scienze Sperimentali Veterinarie, University of Perugia, 06126 Perugia, Italy

**Keywords:** cattle breeds, consumer choice, labeling infographics, meat quality

## Abstract

This work explores how cattle breed influences the qualities of beef that are most important to consumers: color, marbling (which is the fat within the meat), and tenderness. By testing six widely consumed cattle breeds, we found distinct differences such as Angus being the most marbled and tender (*p* < 0.05), while Chianina beef was much brighter in color (*p* < 0.05). We propose that clear visuals and simple labeling could help shoppers to easily understand these traits, making it easier to choose beef that suits their needs. Our findings indicate that improved label transparency, with breed-specific information, empowers consumers to make confident and informed choices, and supports fairer income for producers who supply higher quality meat. This approach could improve satisfaction and trust among both consumers and the beef industry as a whole.

## 1. Introduction

Beef quality remains a central factor influencing consumer satisfaction and purchasing behavior. Modern consumers demonstrate heightened awareness of quality attributes, showing preference for meat that is nutritious, hygienic, and possesses excellent sensory characteristics such as tenderness, juiciness, and flavor. At the same time, labeling and transparency play an increasingly critical role in shaping consumer perception. According to EU Regulation No. 1169/2011, food information must ensure high levels of consumer protection and enable informed decision-making while preventing misleading practices. Consequently, meat labeling can serve not only as a regulatory requirement, but also as a strategic communication tool, fostering trust between producers and consumers when supported by validated, objective data [1,2].

Current beef labeling systems predominantly emphasize origin, traceability, or production practices, yet seldom include scientifically validated information about traits that most influence eating quality, such as tenderness, marbling, or color. The absence of uniform, objective reference values for these indicators limits consumer capacity to compare products meaningfully and may undermine confidence in labeling [3,4,5]. By incorporating standardized, instrumentally measured parameters, it would be possible to provide consistent and transparent information aligned with both consumer interests and regulatory expectations. Studies further indicate that objective, science-based labeling contributes to consumer trust, perceived product value, and overall satisfaction with meat purchases [6,7,8].

Visual attributes such as color, marbling, and tenderness strongly shape consumer expectations and satisfaction levels, often determining repeat purchasing behavior. Breed represents a major biological determinant of these traits, influencing muscle morphology, fat distribution, and biochemical composition. However, the extent of these effects remains insufficiently quantified under harmonized conditions across multiple breeds. The objective assessment of breed differences may provide an important foundation for labeling initiatives aimed at transparency and market differentiation [9,10,11,12].

Multiple aspects of beef quality can be instrumentally measured, including colorimetry, marbling quantification, and textural analysis, which allow more accurate classification and provide data independent of subjective evaluations. While trained sensory panels remain valuable, their use is limited by variability and cost. Objective methodologies such as colorimetric measurement, intramuscular fat quantification, and shear force assessment (e.g., the Warner–Bratzler test) enable reproducible classification and create potential benchmarks for standardized labeling [13,14,15,16].

Marbling, defined as visible intramuscular fat, contributes substantially to the visual appeal and sensory properties of meat, enhancing tenderness, juiciness, and flavor. It is typically evaluated in meat cuts collected between the 5th and 13th ribs and varies among breeds, reflecting genetic and metabolic differences [14,17,18,19]. High marbling content is correlated with richer flavor and improved tenderness perception, though variability remains a challenge in quality grading systems. Tenderness represents another crucial attribute, recognized as one of the strongest predictors of consumer satisfaction and willingness to pay premium prices. Objectively measured tenderness, often assessed via mechanical shear-force methods, ensures reliability and comparability across studies [14,18,20].

The integration of regulated and voluntary labeling frameworks enables producers to communicate additional quality-related information to consumers. Although mandatory EU labeling ensures basic transparency concerning origin, ingredients, and nutritional content, voluntary schemes can further indicate production methods or product quality claims, provided these are scientifically substantiated and verified by recognized certification systems. Scientific underpinnings are essential to guarantee accuracy, reliability, and compliance with consumer protection principles [17,19,20].

Emerging labeling strategies increasingly use visual and informative tools such as infographics to communicate complex product attributes in a clear and accessible format. Effective examples can be found in food sectors including chili pepper spiciness scales (Scoville Heat Units) and chocolate bitterness charts, where standardized visual cues help shoppers quickly identify products that match their preferences [21,22,23,24]. A similar approach could be applied to beef labeling by integrating standardized quality indicators—such as color, marbling, and tenderness—into simple, consumer-friendly infographic formats.

Accordingly, the objective of this study was to assess and compare meat quality characteristics among different cattle breeds, focusing on traits with direct consumer relevance such as color, marbling, and tenderness. The study aims to identify objective and measurable parameters that could serve as reliable indicators of beef quality, thereby contributing to the development of labeling infographics that support transparent communication and informed consumer choices.

## 2. Materials and Methods

### 2.1. Experimental Design

Six different bovine breeds were taken into consideration: German Red Pied, Piemontese, Chianina, Angus, Holstein, and a Polish crossbreed.

For each breed, the meat distributor selected animals from a pool of 446 farms located in Germany (85), Italy (230), Poland (34), and Argentina (97) and from a pool of approximately 50.000 animals for German Red Pied (approximately 8000 animals in 85 farms), Piemontese (approx. 8000 animals in 120 farms), Chianina (approx. 1300 animals in 23 farms), Angus (approx. 22,000 animals in 97 farms), Holstein (approx. 11,000 animals in 87 farms), and Polish crossbreed (approx. 1000 animals in 34 farms). The meat distributor selected animals based on a combination of visual assessment, carcass grading systems, and data-driven evaluations. The process typically involved the following key steps:(1)On-farm selection (live animal assessment). Before slaughter, distributors or procurement specialists evaluated live cattle based on:(i)Body Condition Score (BCS), with an Ideal BCS (3–4 on a 5-point scale) to ensure proper fat cover without excess waste;(ii)Conformation and muscularity, with animals having well-developed muscle mass, particularly in the loin, rump, and shoulder;(iii)Mobility and health, without signs of lameness, disease, or stress;(iv)Weight and age, with slaughter targets (500–700 kg live weight, 12–24 months old).(2)At the slaughterhouse (final selection), additional checks were made:(i)Fat cover evaluation to reject excessively lean or overly fat animals;(ii)Conformation grading to assess muscle development using the SEUROP grading system (only S, E, U, and R accepted).(3)Traceability Tags and RFID (Radio Frequency Identification) chips to ensure that animals meet sourcing standards (organic, antibiotic-free, etc.).(4)Genetic and feed data for feeding history and genetic markers for tenderness.(5)Post slaughter (meat selection), carcasses were first classified according to the SEUROP system for conformation and fat cover, and subsequently evaluated for the specific parameters proposed in this study—color, tenderness, and marbling—by trained assessors within the distributor’s quality program.

From the distributor, an Italian subsidiary of a major German Horeca (hotellerie–restaurant–café) company, we received meat cuts from six different bovine breeds, each sourced exclusively from its country of origin. Specifically, German Red Pied animals (*n* = 63, in 7 shipments) were obtained from Germany; Piemontese (*n* = 117, in 13 shipments), Chianina (*n* = 63, in 7 shipments), and Holstein (*n* = 18, in 2 shipments) were sourced from Italy; Angus (*n* = 63, in 7 shipments) were sourced from Argentina; and Polish crossbreed (*n* = 63, in 7 shipments) were obtained from Poland. In total, 387 animals were included in the study, ensuring that each breed was represented by animals originating entirely from a single country. Each breed was processed in facilities within its respective country of origin, consistent with both legal requirements and established commercial practice. All animals involved in the study were therefore (see point 1, iv above) uncastrated male animals aged from 12 months to less than 24 months with a live weight of between 500 and 700 kg (category A according to the European classification).

From each animal (*n* = 387), three steaks were obtained (for a total of 1161 steaks) and each steak was analyzed in triplicate. The muscle taken into consideration was the Longissimus thoracis et lumborum (LTL), corresponding to the thoracic portion (commonly designated as Longissimus thoracis) sampled between the 7th and 13th ribs. The commercial name of the cut of meat is entrecôte, or boneless rib, obtained from the last six ribs of the loin. Each cut was subjected to analysis to determine color, tenderness, and marbling as described below. The time elapsed between slaughter and cutting was standardized to approximately 4 days, corresponding to the distributor’s logistical and quality control protocols, with minor variations inherent to commercial processing yet remaining within regulatory limits. All samples were skin-packed after cutting and sent to the laboratory. On arrival, the samples were stored at refrigeration temperature (0–4 °C) in the dark for a controlled storage period of approximately 10 days between packaging and analysis, except for the Angus samples (Table 1), which were imported chilled from Argentina by sea freight under a continuous cold-chain system compliant with EU import standards. Due to the longer transport time, Angus meat underwent an extended period of vacuum maturation, which is an intrinsic characteristic of its commercial logistics and contributes to its distinct physicochemical and sensory profile.

### 2.2. Colorimetric Analysis

The «ColourMeter RGB Colourimeter» app (White Marten GmbH, Stuttgart, Germany) was used to measure the color of the samples using an iPhone XS running iOS 13.7, as described by the authors in previously published works [25,26,27]. Before color measurements, the samples were exposed to atmospheric oxygen for 30 min to allow color stabilization. The app enables the user to define the area of the surface to be measured; in our case, a square area of approximately 5 × 5 cm was selected on each steak. This choice was made to record the color as perceived by an observer, rather than the absolute redness of the meat. Consumer perception of color is influenced by multiple visual elements, including marbling, the distribution of lean and fat tissues, and the surface gloss. Therefore, acquiring the average color of a representative portion provides a realistic approximation of the overall impression a consumer would receive when visually assessing the steak. To further standardize this perspective, the smartphone camera was positioned at a 45° angle to the surface of the sample, which approximates the typical viewing angle of a customer examining meat in supermarket conditions.

The «ColourMeter RGB Colourimeter» app was calibrated against a reference colorimeter, a Minolta CR 200 Chroma Meter (Konica Minolta Inc., Tokyo, Japan). Briefly, the Minolta CR 200 Chroma Meter was used to measure a series of red/reddish calibration plates (specifically, the CR-A47 DP, CR-A47 R, and CR-A47 B) in conjunction with a standard white plate in order to determine the CIELAB L* (lightness), a* (redness), and b* (yellowness) colour spaces, and the results were used to calibrate the readout of the «ColourMeter RGB Colourimeter» app. The Minolta CR 200 Chroma Meter was set to measure under the CIE Standard Illuminant D65. D65 is approximately equivalent to the average midday light in Western Europe/Northern Europe, which includes both direct sunlight and diffused light from a clear sky. Hence, it is also referred to as a daylight illumination and has a correlated color temperature of approximately 6500 K. The light used to illuminate the calibration plates for the «ColourMeter RGB Colourimeter» app was, therefore, a source of 6500 K light (Godox Led 64, Godox, Shenzhen, China) under controlled conditions in a photographic light box. The CIELAB system describes colors visible to the human eye according to their hue and chroma (position on the a* and b* axes) and their lightness, L*, which corresponds to a position on a black-to-white scale.

### 2.3. Marbling

To determine the marbling of the samples, the cuts of meat were photographed in a professional photographic setup illuminated with 6500 K LEDs. The camera (Nikon D850, Nikok, Tokyo, Japan) was mounted on a fixed stand to ensure a standard distance between the objective and the sample. The images thus obtained were processed using Adobe Photoshop CS 6 (version 13) on a MacBook Pro Mid 2012 (2.7 GHz Intel Core 7 with card NVIDIA GeForce GT 650 M 1 GB graphics). A square area of 750 × 750 pixels was selected in each image and the number of white pixels was calculated. The percentage of white pixels in this square area provided an indication of the quantity of visible infiltrated fat. This approach is similar to many methodologies of computerized image analysis for the marbling classification described in the scientific literature [28,29,30].

### 2.4. Lipid Extraction and Fatty Acid Analysis

An FAME (fatty acid methyl ester) standard mixture, the SupelcoTM 37-component FAME, was obtained from Supelco (Bellefonte, PA, USA). All solvents and reagents were of analytical grade and were purchased from Carlo Erba Reagents (Milan, Italy). The production of deionized water (>18 MW cm resistivity) was carried out by a Milli-Q SP Reagent Water System (Millipore, Bedford, MA, USA). Total lipids were extracted from meat samples with chloroform/methanol (2:1, *v*/*v*) [31] and subjected to methanolic transesterification using 2N methanolic KOH and hexane [32]. The obtained FAMEs were analyzed by HRGC, using a DANI 1000DPC gas chromatograph (Norwalk, CT, USA) equipped with a split–splitless injector and a flame ionization detector (FID). The separation of FAMEs was achieved by a fused silica WCOT capillary column CP-Select CB for FAMEs (50 m × 0.25 mm i.d., 0.25 μm f.t.; Varian, Superchrom, Milan, Italy), while Clarity integration software (DataApex Ltd., Prague, Czech Republic) was used for acquiring and processing the chromatographic data. The oven temperature was held at 160 °C for 2 min, and then was programmed at 4 °C/min to 225 °C and held for 5 min. The injector and the FID temperatures were set at 250 °C. The carrier gas [33] flow rate was set at 1 mL/min. Fatty acids were identified by comparing the retention times of their methyl esters with the standard FAME mixture, the Supelco™ 37-component FAME. The analytical precision was verified as reported by Piccinetti et al. [34]. The fatty acid compositions were expressed as weight percent (area normalization).

### 2.5. Tenderness

Tenderness was measured on raw meat samples using a Sauter FL 100 digital dynamometer (Sauter Italia, Cinisello Balsamo, Milan, Italy) mounted on a test bench designed for traction and compression measurements and equipped with a digital caliper (Sauter Italia, Milan). The applied methodology was a modified version of the Warner–Bratzler method, the most commonly used system for the instrumental determination of meat tenderness. Briefly, from each slice of meat, six cubes of approximately 1.5 cm^2^ were collected using a hand-held coring device oriented parallel to the longitudinal direction of the muscle fibers. Three cubes were used to measure the resistance to compressive force (applied using a flat-head probe), while the remaining three were tested for resistance to shear force using a wedge-shaped probe; in both cases, the test speed was 250 mm min^−1^. All forces were applied perpendicularly to the fiber orientation. The resulting force–deformation curves were digitally recorded, and the peak force value for each sample was extracted and used for statistical analysis. This protocol, which has also been validated for raw meat in previous studies [35], allowed the assessment of the intrinsic mechanical properties of the muscle without confounding effects due to cooking.

### 2.6. Statistical Analysis

The study employed a completely randomized design (CRD). Animals from each of the six cattle breeds (German Red Pied, Piemontese, Chianina, Angus, Holstein, and Polish crossbreed) were randomly selected as independent experimental units. No blocking factors or repeated measures were incorporated, ensuring all measurements (color, marbling, tenderness) were taken under identical conditions per animal.

Statistical analysis was performed using the Kruskal–Wallis test to compare differences among cattle breeds, followed by Dunn’s multiple comparisons. Analyses were conducted in GraphPad Prism 8.4.3 (GraphPad Software, Boston, MA, USA). Dunn’s test presents adjusted *p* values for each pairwise comparison, indicating which breed pairs differ significantly. The graphical display of the results was obtained with Prism 8.4.3 software for Mac OS.

### 2.7. Labeling System

Data obtained from this work (Table 2) were transformed into a 5-level scale using the formula below, using the reciprocal for tenderness values (where the lower the value, the higher the score):x = v_p_ * 5/v_max_
where

x: 5-level scale value;

v_p_: value for any given parameter;

v_max_: the highest values for any given parameter.

The scores were then converted into a simple infographic (see graphical abstract).

**Table 2 animals-15-03467-t002:** Results of colorimetric, tenderness, marbling, and total lipids evaluations, expressed as mean and standard error (se).

	*L**		*a**		*b**	
	Mean	se	Mean	se	Mean	se
Angus	32.33 ^b^	0.38	19.46 ^c^	0.17	17.54 ^c^	0.26
Chianina	34.79 ^b^	0.67	26.64 ^a^	0.38	23.49 ^b^	0.41
Holstein	33.78 ^b^	0.3	22.44 ^b^	0.48	21.00 ^bd^	0.46
German Red Pied	27.64 ^a^	0.41	19.71 ^c^	0.29	17.57 ^c^	0.33
Piemontese	26.39 ^a^	0.29	19.55 ^c^	0.18	15.59 ^a^	0.23
Polish crossbreed	31.89 ^b^	0.37	21.37 ^b^	0.27	18.92 ^cd^	0.29
	Tenderness (Compression) (Newton, N)			Tenderness (Shear Force) (Newton, N)		
	Mean		se	Mean		se
Angus	48.04 ^a^		2.94	15.42 ^b^		0.77
Chianina	66.18 ^b^		2.72	15.24 ^b^		0.82
Holstein	43.44 ^ac^		4.42	15.26 ^ab^		1.56
German Red Pied	53.35 ^acd^		2.24	12.98 ^b^		0.65
Piemontese	59.23 ^bc^		2.05	15.22 ^b^		0.62
Polish crossbreed	63.23 ^bd^		2.39	21.62 ^a^		1.12
	Marbling (% on the Total Surface of the Sample)			Total Lipids (% of the Total Sample Weight)		
	Mean		se	Mean		se
Angus	27.01 ^b^		1.52	5.35 ^bc^		0.50
Chianina	14.60 ^cd^		0.79	6.30 ^b^		0.37
Holstein	17.51 ^bc^		1.22	5.81 ^b^		0.58
German Red Pied	11.80 ^ad^		0.62	3.09 ^ac^		0.13
Piemontese	10.23 ^a^		0.37	2.63 ^a^		0.08
Polish crossbreed	18.83 ^b^		0.72	4.81 ^b^		0.31

*L**, *a**, *b**: colorimetric coordinates, *L** (lightness) represents the brightness of the color, ranging from 0 (black) to 100 (white); *a** is the red–green axis, where positive values indicate red and negative values indicate green; *b** is the blue–yellow axis, where positive values indicate yellow and negative values indicate blue. Different letters in the same column indicate means with statistically significant differences (*p* < 0.05).

## 3. Results

### 3.1. Colorimetric Analysis

The results of the colorimetric analysis are reported in Table 2. The colorimetric analysis highlighted how meat of the Chianina, Holstein, and Polish crossbreed breeds are significantly redder than the others, showing higher *a** coordinate values (26.64 ± 3.02, 22.44 ± 2.04, and 21.37 ± 2.11, respectively). Regarding the value of the *b** coordinate, which indicates the tendency toward yellow or cyan (higher values indicate yellow, lower values indicate cyan), the same three breeds (Chianina, Holstein, and Polish crossbreed) showed higher values (23.49 ± 3.24, 21.00 ± 1.94, and 18.92 ± 2.33, respectively). Finally, the meat of the Chianina (34.79 ± 5.30), Angus (32.33 ± 3.05), Holstein (33.78 ± 1.26), and Polish crossbreed (31.89 ± 2.95) appeared brighter, showing higher *L** coordinate values, especially when compared with German Red Pied (27.64 ± 3.24) and Piemontese (26.39 ± 3.18), with a darker shade.

### 3.2. Marbling, Total Lipids, and Fatty Acid Profile

The results of the marbling and total lipid evaluation are reported in Table 2. The meat of the Angus (27.01 ± 12.09) was the most marbled, followed by the Polish crossbreed (18.83 ± 5.75), Holstein (17.51 ± 5.18), and Chianina (14.60 ± 6.28), and finally by German Red Pied (11.80 ± 4.96) and Piemontese (10.23 ± 4.05), with statistically significant differences. As for the total lipid content, expressed as a % of the total weight of the sample, the highest value was detected for Chianina (6.30 ± 1.94), followed by Holstein (5.81 ± 2.48) and Angus (5.35 ± 2.61), and then by Polish crossbreed (4.81 ± 1.62), and finally by German Red Pied (3.09 ± 0.67) and Piemontese (2.63 ± 0.44). Regarding the total lipid content, the statistical analysis highlighted a good correlation with the visible marbling (Figure 1). The results of the analysis of the fatty acid profile are reported in Table 3. Differences, although not always statistically significant, were detected in the content of omega 3 fatty acids, with higher percentages in Piemontese and German Red Pied breeds and lower levels in Chianina and Holstein breeds, with the Angus at an intermediate level. Concerning omega 6 fatty acids, the highest content was detected in Polish crossbreed, followed by Holstein, Piemontese, Chianina, Angus, and finally German Red Pied. In general, saturated fats—in particular, myristic, palmitic, and stearic fatty acids—represented 45–48% of the total lipids, monounsaturated fats represented 35–45%, and polyunsaturated fats 5%. The most represented polyunsaturates were linoleic and linolenic acid, while the monounsaturates were mainly represented by oleic acid.

### 3.3. Tenderness

The results for tenderness are reported in Table 2. The lowest value of resistance to the application of a compressive force was recorded for Holstein (43.44 ± 18.75 N), followed by Angus (48.04 ± 23.30 N), German Red Pied (53.35 ± 17.77 N), Piemontese (59.23 ± 22.17 N), Polish crossbreed (63.23 ± 18.71 N), and Chianina (66.18 ± 21.61). With average values of 43.44 and 48.04 N, the meat of Angus and Holstein breeds were found, with statistically significant differences, to be the most tender in the compression test, while, with values of 63.23 and 66.18 N, respectively, the meat from the Polish crossbreed and Chianina breeds was the most consistent. Intermediate values were recorded for German Red Pied and Piemontese. Similar, but more leveled, results were highlighted in the shear force test. The lowest value was recorded for German Red Pied (12.98 ± 5.16 N), followed by Piemontese (15.22 ± 6.69 N), Chianina (15.24 ± 6.49 N), Holstein (15.26 ± 6.63 N), Angus (15.42 ± 6.12 N), and Polish crossbreed (21.62 ± 8.86).

## 4. Discussion

The relationship between breed and beef quality has long been discussed. Many different breed-dependent factors have been positively or negatively associated with beef quality, such as age at physiological maturity; growth path; muscle structure; amount, composition, and distribution of intramuscular fat; and content of connective tissue [36]. A well-known example is the Wagyu breed, which produces meat characterized by intense marbling, flavor, and juiciness [37]. Genetic selection for cattle has led to a clear differentiation between meat and dairy breeds, with meat-producing breeds often associated with higher-quality meat when compared to dairy breeds. The age of physiological maturity is another important breed-dependent parameter that strongly influences meat quality: in fact, the deposition of intramuscular fact starts after the deposition of subcutaneous fat, when the animal matures. Therefore, early-maturing breeds such as Angus show higher levels of intramuscular fat than breeds that mature later when slaughtered at the same age [33]. This study demonstrates that breed has a significant impact on key beef quality traits, specifically color, marbling, and tenderness. The observed differences among breeds are not only statistically significant, but also biologically meaningful, reflecting underlying genetic and physiological mechanisms that govern muscle development, fat deposition, and meat quality.

The higher intramuscular fat content and marbling observed in Angus samples compared to Holstein and Piemontese are consistent with the breed’s earlier physiological maturity and genetic predisposition for fat deposition. Early-maturing breeds such as Angus tend to deposit intramuscular fat sooner, resulting in greater marbling and, consequently, enhanced juiciness and flavor. In contrast, later-maturing breeds may produce leaner meat when slaughtered at the same age, as intramuscular fat deposition occurs later in their growth trajectory. These findings align with previous research highlighting the role of breed in determining the timing and extent of fat accumulation within muscle tissue [38,39,40,41,42]. The fatty acid profiles further illustrate breed-specific differences. Angus beef exhibited a higher proportion of oleic acid (C18:1n-9 + n7) and a lower proportion of linoleic acid (C18:2n-6) compared to Holstein, while Piemontese animals had a notably lower C18:2n-6 content than reported in the literature. These variations can be attributed to genetic factors influencing the activity of enzymes, such as stearoyl-CoA desaturase, that modulate the synthesis and composition of fatty acids in muscle. The higher monounsaturated fatty acid content in Angus may contribute not only to marbling, but also to the palatability and nutritional profile of the meat [38,39,40,42,43,44,45]. The amount of fat in a meat sample is determined by chemical methods but this approach measures more than the intramuscular fat visible to the human eye. Many studies have investigated the relationship between marbling perceived by the human eye and the amount of fat in the sample, generally reporting a good correlation between the two even if the magnitude of this correlation is debated [2,28,39,46,47,48]. Such variability could be explained on the basis of different chemical extraction methods for lipids as well as different methods to assess the amount of visible fat [46]. The good correlation observed in our study is consistent with a study by Konarska et al. [49], which reported a moderate correlation between visible marbling and intramuscular fat percentage [49]. The fatty acid profile of meat can vary between different cattle breeds due to genetic, physiological, and environmental factors [16]. Phospholipids represent a higher proportion of total fat in breeds with a lower amount of lipids in the muscle; therefore, these breeds usually have higher proportions of polyunsaturated fatty acids (PUFAs) [50]. Differences in the portioning of body fat between dairy and beef breeds are also reported in the literature, with dairy breeds having more non-visible fat and less subcutaneous fat [51]. The results obtained in this study corroborate this thesis, considering that Holstein showed the second-highest values for total lipids. In a study by Warren et al. [51], the beef fatty acid composition of Aberdeen Angus cross and Holstein–Friesian breeds was compared. The study reported that the amounts of neutral lipids and phospholipids in the *longissimus* muscle were not very different and the proportions of many fatty acids were also quite similar. In our study, while the total amount of lipids was similar between Angus and Holstein, differences were observed in the fatty acid profile: in particular, the Angus samples showed higher concentrations of C18:1n-9 + n7 (35.9% vs. 27.7%) and lower concentrations of C18:2n-6 (7.9% vs. 15.7%). With regard to Holstein and Piemontese, the concentration of the major fatty acids (palmitic, stearic, oleic, and linoleic) expressed as percentage of the total found in our study was consistent with other works. It is worth noting the percentage of C18:2n-6 in Piemontese animals, which, in our study, is lower than reported in the literature (10.2% compared to 16.8% and 27.4%) [52]. The only statistically significant differences were observed for C17:1 and C18:1 n9 + n7 between German Red Pied (higher values) and Holstein (lower values) in the Kruskal–Wallis test, followed by Dunn’s multiple comparisons (*p* < 0.05).

Tenderness, a critical determinant of consumer satisfaction, also varied among breeds. The use of a modified Warner–Bratzler method allowed for the assessment of tenderness in the raw meat, reflecting the characteristics available to consumers at the point of purchase. The variability in tenderness across breeds can be explained by differences in muscle fiber composition and connective tissue content, both of which are influenced by genetic selection for meat or dairy production. Breeds selected for meat production often exhibit muscle structures that favor greater tenderness, whereas dairy breeds may have higher connective tissue content, resulting in firmer meat [20,42,53,54]. The strong correlation observed between visually assessed marbling and chemically measured intramuscular fat supports the validity of using visual cues as practical indicators of eating quality for consumers. However, the strength of this relationship can be affected by the methods used for fat extraction and visual assessment, as noted in previous studies [47,54,55]. Tenderness is considered the single most important factor for consumer’s perception of beef quality. It is widely accepted that beef tenderness is a rather inconsistent characteristic, with huge variation between breeds, animals, meat cut, and many other variables. This inconsistency is recognized as one of the main problems for the beef industry around the world [56]. The variability of beef tenderness evaluation is also linked to the difficulty in the development of a standardized, repeatable, and reliable method to objectively assess this characteristic at the laboratory level [57]. Researchers have systematically attempted to determine the most repeatable and accurate method to assess beef tenderness. To date, the Warner–Bratzler method and its variations are recognized as the best approach to measure beef tenderness [58]. For the purpose of the present study, a modified version of the Warner–Bratzler method was used. In particular, samples were not cooked before tenderness measurement. The cooking method, temperature, and time for meat samples in the original methodology can be modified depending on the hypothesis investigated in each experiment and impact the final results [20]. The aim of the present study was to classify beef on the basis of characteristics that can be evaluated at the time of purchase and to hypothesize a series of infographics for the label. Therefore, cooking is a process that happens later, depends on the consumer’s preference, and is independent of the original beef characteristics. Other examples of applications of modified versions of the Warner–Bratzler method on non-cooked beef can be found in the literature [59,60].

Visual appearance is the first level of beef quality perceived by the consumer. Meat color greatly influences the visual appearance of beef as well as the consumer’s choice. Bright red beef is associated with freshness and higher quality, while paler or darker beef is often perceived to be near spoilage or of lower quality [61]. The red color of meat is due to the conversion of deoxymyoglobin into oxymyoglobin, which has a red color, after exposure to oxygen. Prolonged exposure to oxygen activates oxidative metabolism and leads to the accumulation of free radical by-products, which are responsible for the oxidation of myoglobin into metmyoglobin and the consequent brown coloration of meat [62,63,64]. Studies report that the majority of the variability observed in beef color is related to L* coordinates in conjunction with the a* coordinates, accounting for the oxidation state and pigmentation [61,63]. On the other hand, the level of intramuscular fat content and the redox state is more related to the b* coordinate [16].

These breed-specific differences in meat quality traits provide a scientific foundation for the development of an objective labeling model. The proposed labeling system, based on standardized, instrumentally measured attributes such as color, marbling, and tenderness, addresses current gaps in consumer information. By translating complex biological variability into clear, actionable information, the labeling model enables consumers to make more informed choices according to their preferences for attributes like tenderness or marbling. This approach also supports transparency and fair competition among producers, as it rewards the production of higher-quality beef based on measurable criteria rather than subjective claims. The integration of an infographic or grading scale into beef labeling has practical implications for both consumers and the industry. Visual tools can distil complex data into accessible formats, helping consumers to quickly identify products that meet their needs and preferences at the point of sale. Such a system aligns with regulatory goals for accuracy and transparency in food information, while also enhancing consumer trust and satisfaction. The usefulness of an infographic in this context cannot be overstated. By visually presenting complex information, such as breed characteristics, fat content, tenderness, and marbling, an infographic can simplify these factors for the consumer. This visual guide can help consumers to make more informed decisions at the point of purchase, allowing them to understand the key traits of the beef they are considering, such as its potential tenderness or marbling. With this tool, consumers can easily navigate the often overwhelming array of beef options, making their purchasing decisions more straightforward and based on clear, easily digestible information [65].

## 5. Conclusions

This study demonstrated that cattle breed significantly influences key beef quality attributes—color, marbling, and tenderness—each shaping consumer perception and preference. Among the six breeds analyzed, Angus showed higher marbling and tenderness, whereas Chianina exhibited the brightest meat color, confirming breed-specific effects on meat quality. The proposed labeling model, grounded in standardized and instrumentally measured parameters, provides a transparent and reliable framework for communicating such complex quality information. Integrating visual tools like infographics can effectively bridge scientific data and consumer understanding, facilitating informed choices and supporting the regulatory goals of accuracy and transparency. By translating biological diversity into actionable consumer information, this approach enhances trust in beef products, supports fair market competition based on measurable quality traits, and contributes to a more transparent and satisfying consumer experience.

## Figures and Tables

**Figure 1 animals-15-03467-f001:**
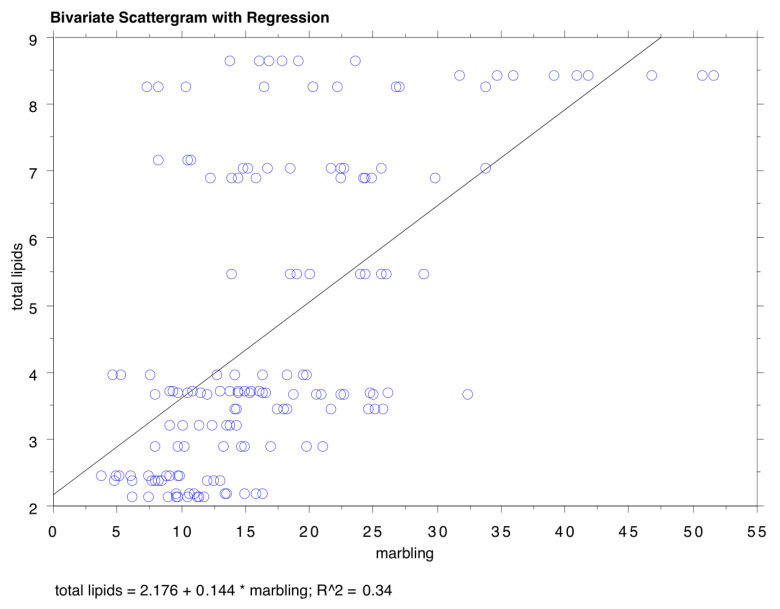
Bivariate scattergram with regression plot of marbling and total lipids. Total lipids = 2.176 + 0.144 * marbling; R = 0.583; R^2^ = 0.34.

**Table 1 animals-15-03467-t001:** Time from slaughter to analysis (days).

	From Slaughter to Cutting/Packaging		From Cutting/ Packaging to Arrival at the Lab		From Arrival to Analysis		From Packaging to Analysis	
	Mean	se	Mean	se	Mean	se	Mean	se
Angus (*n* = 7 shipments)	4.29	0.64	26.71	16.64	9	1	35.71	16.22
Chianina (*n* = 7 shipments)	5	0.44	0	0	9	0	9	0.00
German Red Pied (*n* = 7 shipments)	3.86	0.63	3.43	0.53	7.71	0.18	11.14	0.51
Piemontese (*n* = 13 shipments)	3.77	0.23	3.08	0.4	6.92	0.51	10	0.32
Polish crossbreed (*n* = 7 shipments)	3.71	0.47	0.57	0.2	12	0	12.57	0.27
Holstein (*n* = 2 shipments)	4	0	1	0	7.5	3.5	8.5	3.5

**Table 3 animals-15-03467-t003:** Fatty acid composition. Results are expressed as mean and standard error (se).

	Angus		Chianina		German Red Pied		Piemontese		Polish Crossbreed		Holstein	
	Mean	se	Mean	se	Mean	se	Mean	se	Mean	se	Mean	se
C14:0	1.81	0.08	2.11	0.09	1.27	0.03	1.91	0.10	2.17	0.03	2.64	0.07
C14:1	0.37	0.05	0.45	0.01	0.25	0.01	0.49	0.02	0.33	0.01	0.36	0.00
C15:0	0.44	0.04	0.7	0.03	0.44	0.02	0.43	0.03	0.33	0.02	0.45	0.00
C15:1	0.17	0.01	0.26	0.02	0.16	0.00	0.17	0.01	0.15	0.01	0.16	0.00
C16:0	24.3	0.24	24.7	0.90	22.7	0.29	25	0.54	25.4	0.43	25.9	0.10
C16:1n-9	0.26	0.01	0.28	0.00	0.24	0.00	0.23	0.02	0.17	0.01	0.28	0.01
C16:1n-7	3.21	0.14	2.96	0.08	3.09	0.02	4.29	0.10	3.13	0.03	2.82	0.07
C17:0	0.97	0.04	1.26	0.05	0.93	0.03	0.75	0.03	0.69	0.06	0.79	0.01
C17:1	1.11 ^ab^	0.03	1.19 ^ab^	0.05	1.62 ^b^	0.02	1.02 ^ab^	0.01	0.79 ^ab^	0.06	0.62 ^a^	0.02
C18:0	15.8	0.44	18.2	0.37	14.4	0.21	12.8	0.22	12.5	0.30	16.6	0.14
C18:1 trans	1.19	0.14	0.96	0.10	0.83	0.04	0.73	0.03	0.71	0.04	1.15	0.07
C18:1n-9 + n7	35.9 ^ab^	0.95	33.3 ^ab^	0.47	40 ^b^	0.56	33.5 ^ab^	0.83	30.5 ^ab^	0.34	27.7 ^a^	0.17
C18:2n6	7.87	0.23	10.1	0.89	6.02	0.30	10.2	0.62	13.8	0.67	15.7	0.27
C18:3n3	1.21	0.14	0.59	0.03	1.59	0.12	0.92	0.07	0.88	0.14	0.71	0.04
9c, 11t CLA	0.28	0.01	0.24	0.03	0.22	0.00	0.21	0.01	0.15	0.00	0.34	0.03
C20:1	0.23	0.01	0.23	0.03	0.22	0.01	0.18	0.01	0.13	0.00	0.12	0.01
20:3n6	0.86	0.03	0.61	0.01	0.89	0.07	1.14	0.09	1.64	0.18	0.55	0.02
C20:4n6	2.62	0.20	1.57	0.13	2.43	0.25	3.73	0.46	5.56	0.56	2.56	0.05
C20:5 n3	0.53	0.07	0.08	0.01	0.85	0.11	0.67	0.08	0.24	0.03	0.12	0.01
C22:5n3	0.86	0.11	0.2	0.03	1.51	0.18	1.44	0.23	0.7	0.06	0.4	0.03
C22:6n3	0.05	0.01	0.01	0.00	0.35	0.06	0.22	0.03	0.05	0.01	0.03	0.01

Different letters in the same row indicate means with statistically significant differences (*p* < 0.05).

## Data Availability

Data provided to the Journal.

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
