# Peer review of "Definition of Meat Quality Across Different Cattle Breeds"

_animals, 2025, doi:10.3390/ani15233467_

Round 1
Reviewer 1 Report
Comments and Suggestions for Authors
The current manuscript presents a study aiming to evaluate different meat parameters (color, tenderness, marbling and fatty acid profile) of six different beef breeds to propose labeling infographic helping consumer to make decision during purchasing. The subject is scientifically interesting, and numbers of samples analyzed in this study is important. However, it needs some improvements.
The title is not adequately presenting the paper content.
The introduction is too long that should be reduced.
Also, results are not clearly represented. In my opinion, the table 1 is not necessary to be added, the information in this table could just be described in the text. The titles of all figures and tables should be reviewed to be more comprehensive and informative. In the tables, it is better if the units were put next to the parameters. The number of samples of each breed should be added in these tables. Authors present too much data in the tables, please keep averages and add standard error of the means and the probability.
For the main text, authors studied fatty acid profile, but they focused more on the three parameters (marbling, tenderness and marbling). Please, in results section, keep just averages without standard error, it will be more clear. In the discussion, authors mention “in the literature”, “many studies” and “other works” without adding references, Please add them. In the line 420, the year of the study is missed.
For the infographic labeling proposed, It should be improved. Color and tenderness pictures are not representative. For color it is more convenient if authors used gradient of color or letter for this parameter, and for tenderness I suggest using letter.
Conclusion section needs to be improved. This section is not representing the manuscript content.
Reviewer 2 Report
Comments and Suggestions for Authors
Title: by quickly scanning, it does not seem you did any testing with consumers, so consumer and labeling does not belong in the title. Please revise.
Abstract: There must be statistical evidence in the abstract. Please use p-values when discussing differences or lack thereof.
Ln33-34: delete first sentence of abstract. This doesn’t belong in an abstract.
Ln34: change paper to study (or trial or experiment). The paper didn’t do anything.
Ln36: how many animals per breed? Were they the same sex? Were they fed the same? Cattle should be managed the same and they should be the same sex to make any meaningful inferences about breed differences or you must have the same variation in breed, management, age, etc in each breed. You will also need sufficient numbers per breed.
Ln47-49: you haven’t provided any insights for this basis. You have provided a fairly oversimplified summary of results without any statistical significance. How does this warrant including breed information on packages? You didn’t ask consumers if they would like to see this information.
Introduction is far too long. Please condense considerably, focusing on what is known about the topic at hand, what is the knowledge gap, what are the objectives of the study to fill the knowledge gap, and what are the hypotheses? The entire first paragraph should be deleted. It adds nothing.
Ln69-70: sure tenderness can’t be known without cooking and eating the product. However, marbling and coloring can be clearly observed by the consumer at the point of purchase, so why is additional information needed on the package.
Ln79-86: how did you go 15+ lines without providing references? Much of this information can probably be eliminated.
Ln87-93: you repeat the sentence and change the words slightly, yet you have different references for each sentence. I’m confused how this is possible. This introduction needs a complete overhaul. Just start from scratch. Don’t try to salvage it. I highly recommend removing reference to labeling, labeling standards. Focus on the actual experiment - Meat quality differences of different breeds of cattle. Why did you select those breeds? What is known about those breeds? What is the missing information about meat quality of those breeds?
Ln161: you can’t hypothesize an infographic. What is the scientific hypothesis you are testing?
Ln165-191: please include summary statistics for all the selection traits, particularly weight, age, fat cover, conformation score by breed.
Ln167: please include a table with breakdown of animals by breed and by country. It appears as though breeds were selected entirely within 1 country (i.e. all angus came from Argentina). This is confounded. Management practices unique to the country could just as easily contribute to breed differences.
Ln167: obviously all animals were not harvested at a common abattoir. This should be stated. What are the “meat distributor’” credentials to make these selection decisions? What training do they have? Was there only 1 meat distributor? How were cattle in Argentina evaluated by the same meat distributor??
Ln190: how were carcasses graded? You’ve already suggested carcass conformation was conducted. There is not an eating quality grading system utilized by the countries mentioned.
Ln199: were 3 steaks cut from a single muscle sample? This wording is confusing.
Ln203: if taken from the last 6 ribs, there was not any longissimus lumborum (from the lumbar region). It should be referred to as simply longissimus thoracis.
Ln205: if something is standardized, there should not be any variation. So was it 4 days? If all weren’t exactly 4 days, why not?
Ln208: see previous comment. There should not be an approximation for a standardized period of time.
Ln209: why and how were angus handled differently??
Ln216: You need to describe how color measurements were actually taken. You spend the entire section talking about how it was calibrated against an actual colorimeter, so I’m confused why you didn’t simply use the colorimeter. How long were samples exposed to oxygen before taking color readings. Given this seems to be a relatively new technology and not globally available, I suggest a figure (photo) demonstrating how the color readings were obtained. Aperture size and observer angle are also standard information that should be provided when collecting and presenting color data.
Ln240-250: why wasn’t a trained professional used to assess marbling?
Ln273-286: were samples cooked prior to tenderness testing?? If so, please describe cooking process (type of cooking device, target internal temperature, how was temperature monitored). If samples were not cooked, there is very little useful information here.
Ln288: how did you account for the additional aging period of Angus? They were aged at least 24 days longer than the other breeds before analysis, which would undoubtedly improve tenderness.
Table 2: please include sample size – ideally total and by breed
Table 2: superscripts must be checked for accuracy. There are some traits where it seems there is no rhyme or reason how the superscripts were assigned. You should start with the highest or lowest mean (typically it’s highest) and assign “a” and go from there. Superscripts shouldn’t have a gap. Here are a few examples (there are more): Holstein b*: bd, German Red Pied tenderness: acd, German Red Pied marbling: ad. It’s next to impossible to decipher results and make sense of the results as written because the superscripts don’t make sense.
Ln324-333: Where’s the statistical evidence that there were differences. Please include p-values. This is a critical omission. Also, please don’t include values from the table in the text. Otherwise, it defeats the purpose of having a table. Reference the table and the reader can see all of the values. Chianina had greater a* values than all other breeds, so the way you’re reporting the results is not accurate. Piedmontese actually had greater b* values than all other breeds. How does Holstein have bd superscript?? This doesn’t make sense.
Ln331: and polish crossbred
Discussion: Given the lack of information in methodology about selection of animals for inclusion or how animals didn’t get selected and reporting of simple summary statistics, I can’t make any further comments on discussion until that issue is resolved. The variation in production practices and ignoring the extended aging of Angus confounds the breed “differences”. Superscripts don’t seem to be assigned appropriately, so what are the actual differences?? The lack of an actual subjective marbling score is another key concern.
Round 2
Reviewer 1 Report
Comments and Suggestions for Authors
Please add Standard Error of the Mean (SEM) instead of the Standard Deviation (SD) for all means in tables. The SEM is more convenable because is shows the means variability. The authors should add the probability for each tested parameters in the tables. Please remove the Minimum (Min) and Maximum (Max) values, because in my opinion they are not necessary. The color indexes in the footnote of table 2 should be explained more (L*: lumonisty, etc.). Figure 2 is confusing. I was wondering if it is showing the results of this study or it is an infographic explaining the meat quality.
Author Response
Dear Editor,
Thank you for the opportunity to revise our manuscript. We have carefully addressed all of the reviewer’s specific recommendations to improve the clarity of our study. The following changes have been made: i) the Standard Error of the Mean (SEM) has been provided for all means in the tables, replacing the previously reported Standard Deviation (SD), to more appropriately represent the variability of the data, ii) or each parameter tested, the probability (p-value) has been added in the respective tables, in order to provide readers with the necessary statistical context. iii) all Minimum (Min) and Maximum (Max) values have been removed from the tables, as suggested, to streamline the presentation and avoid unnecessary detail, iv) the footnote of Table 2 has been expanded to explain all colour indexes in detail for improved clarity, v) to address concerns about Figure 2, we have clarified in the revised manuscript that it is an infographic designed to summarize and communicate the principal meat quality results obtained in this study.
We trust these amendments fully meet the reviewer’s recommendations and enhance the clarity of the manuscript.
Thank you again for your time and consideration.
Sincerely,
Beniamino Cenci Goga
On behalf of all authors
